# LAYER RECURRENT NEURAL NETWORKS

**Weidi Xie, Alison Noble & Andrew Zisserman**
Department of Engineering Science, University of Oxford, UK

## ABSTRACT

In this paper, we propose a Layer-RNN (L-RNN) module that is able to learn contextual information adaptively using *within-layer* recurrence. Our contributions are three-fold: (i) we propose a hybrid neural network architecture that interleaves traditional convolutional layers with L-RNN module for learning long-range dependencies at multiple levels; (ii) we show that a L-RNN module can be seamlessly inserted into any convolutional layer of a pre-trained CNN, and the entire network then fine-tuned, leading to a boost in performance; (iii) we report experiments on the CIFAR-10 classification task, showing that a network with interleaved convolutional layers and L-RNN modules, achieves comparable results (5.39% *top1* error) using only 15 layers and fewer parameters to ResNet-164 (5.46%); and on the PASCAL VOC2012 semantic segmentation task, we show that the performance of a pre-trained FCN network can be boosted by 5% (mean IOU) by simply inserting Layer-RNNs.

## 1 INTRODUCTION

In computer vision tasks, such as image classification or pixel level prediction, multi-scale contextual information plays a very important role in achieving high performance. The original architectures for these tasks (e.g. He et al. (2016a); Krizhevsky et al. (2012); Long et al. (2015); Ronneberger et al. (2015); Simonyan & Zisserman (2015); Szegedy et al. (2015)) were able to obtain multi-scale context with a large spatial footprint by the combination of filters through the layers of the network, so that a large receptive field was effectively built up. Indeed, the final layers of these networks use average pooling or fully connected layers (convolution with a large kernel) so that the effective receptive field covers the entire input image patch. More recent pixel prediction architectures have used dilated convolutions (Yu & Koltun, 2016; Chen et al., 2016) which are able to aggregate multi-scale contextual information without losing resolution (due to the spatial pooling and strides in the original architectures), and without incurring the penalty of having to learn many parameters for convolutions with very large kernels.

In this paper we introduce an alternative 'module' for learning multi-scale spatial contextual information by using Recurrent Neural Networks (RNNs) *within* layers. This approach is inspired by the ReNet architecture of Visin et al. (2015), which we extend here into a hybrid architecture that *interleaves* traditional convolutional neural network (CNN) modules with layer recurrent modules, and we term a Layer Recurrent Neural Network (L-RNN). A L-RNN module is a combination of 1D RNNs, and is able to learn contextual information adaptively, with the effective receptive field able to reach across the entire feature map or image, if that is required for the task. The hybrid network combines the best of both worlds: canonical CNNs are composed of filters that are efficient in capturing features in a *local* region, whilst the L-RNNs are able to learn *long-range* dependencies across a layer efficiently with only a small number of parameters.

We describe the basic L-RNN module in Section 2, and discuss different fusion choices for the hybrid architecture by incorporating L-RNN into residual blocks (He et al., 2016b) in Section 3. In addition, in Section 4, we explain how L-RNN modules can be inserted into pre-trained CNNs seamlessly. This means that the entire network does not have to be trained from scratch, only the added L-RNNs are fine-tuned together with pre-trained networks, and the experiments show that this addition *always* improves performance. In Section 5, we experiment on the CIFAR-10 classification with the hybrid networks of increasing depths, by using Layer Normalization (Ba et al., 2016), we are able to train vanilla RNNs to match the performance of GRU (Chung et al., 2015), while

using fewer parameters. In addition, we fine-tune a truncated VGG-16 FCN base net for semantic segmentation on the Pascal VOC 2012 and COCO (Lin et al., 2014) dataset.

It is worth noting that (broadly) recurrence can be used in feed-forward multi-layer convolutional neural network architectures in two ways: *between* layers, and *within* layers. For example, between-layer recurrence was used for scene labelling in (Liang et al., 2015; Pinheiro & Collobert, 2014) with convolutions applied recursively on top of feature maps from different layers or raw input images. And in (Zheng et al., 2015), spatial dependencies are modelled explicitly for semantic segmentation with densely connected Gaussian CRFs by iterated application of bilateral filtering using between-layer recurrence.

By contrast, our Layer-RNN architecture falls into the second category, where *within-layer* recurrence is used to capture dependencies. Others have learnt contextual information from within layer recurrence for tasks such as object detection (Bell et al., 2016), and low-level vision problems, such as de-noising, colourization and smoothing (Liu et al., 2016). We postpone discussing in detail the relationships of the proposed Layer-RNN modules to these architectures, and to that of ReNet (Visin et al., 2015) and ReSeg (Visin et al., 2016), until we have introduced the L-RNN in Section 2.

## 2 LAYER-RNN ARCHITECTURE

The architecture of the network (Figure 1) is composed of two parts. Local features are calculated by the low-level CNNs module, the Layer-RNN (L-RNN) module, consisting of several 1D spatial RNNs is applied to capture the spatial dependencies. By scanning across the feature maps in different directions, the complete L-RNN is able to learn the receptive field in an adaptive way, up to the size of the entire image. These two modules can be combined to build networks in various ways; for example, an L-RNN module can be stacked on top of several CNN modules at the final layer, or CNN and L-RNN modules can be interleaved at multiple levels.

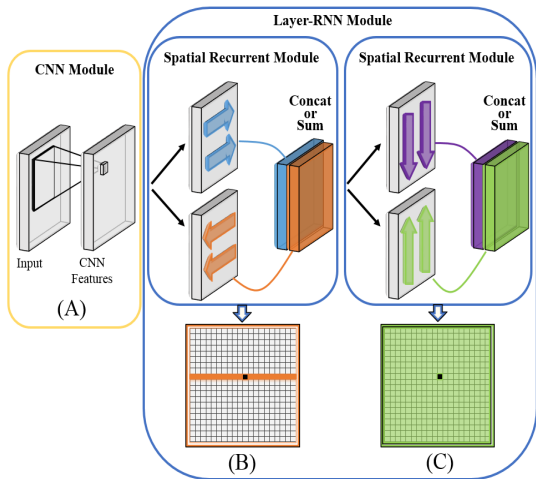

Figure 1: **Basic Architecture:**
Given the input image, local features are calculated by the CNN module (A).
In (B), two 1D spatial RNNs are applied to scan along each row independently from different directions, hidden states are calculated at every spatial step, and the output feature maps can either be concatenated or summed up. The receptive field for the black pixel in (B) is labelled in orange;
In (C), two 1D spatial RNNs are applied to scan along each column from two directions. The combination of (B) and (C) defines the L-RNN module that is able to propagate information over the entire image.

### 2.1 LAYER-RNN MODULE

As shown in Figure 1, the Layer-RNN (L-RNN) module is a combination of the 1D spatial recurrent modules (B) and (C). In each module, there are two 1D RNNs scanning across the feature maps horizontally or vertically from two directions (bidirectional spatial RNNs), and their hidden states are updated at every spatial step. Consequently, for each of the horizontal and vertical directions, two output feature maps are obtained with the same width and height as the input feature maps. In our implementation, we simply sum up these output feature maps (an alternative is to concatenate the output feature maps, but that would increase the number of parameters).

More formally, assume the feature maps (layer $L$) coming into the L-RNN module are $X^L \in \mathbb{R}^{m \times n \times d}$ and output $X^{L+1}$ (layer $L + 1$), where $m, n, d$ refers to the width, height, and the number of feature maps respectively for the input layer. For simplicity, assume the input to the 1D spatial

RNNs from $X^L$ is a feature vector at each spatial location, each row or column on the feature maps is treated as one sequence. When scanning from left to right, the feature responses for location $ij$ can be calculated:

$$x_{i,j}^{L+1} = f(Ux_{i,j}^L + Vx_{i,j-1}^{L+1} + b) \qquad \text{left to right} \qquad (1)$$

Where $x_{i,0}^{L+1} = 0$, $x_{i,j}^L \in \mathbb{R}^{d \times 1}$, $x_{i,j}^{L+1}, x_{i,j-1}^{L+1} \in \mathbb{R}^{D \times 1}$, $U \in \mathbb{R}^{D \times d}$, $V \in \mathbb{R}^{D \times D}$, $b \in \mathbb{R}^{D \times 1}$, $D$ denotes the number of nodes used in the 1D spatial RNN, and $f$ refers to the non-linearity function. 1D spatial RNNs scanning other directions can be calculated similarly. Notice that, the first term of equation 1 encodes local information independently, resembling the normal convolutional layer, and the second term characterizes the *within-layer* recurrence ($U$ is a convolution matrix, $V$ a recurrence matrix). We make use of this observation in Section 4.

## 2.2 DISCUSSION AND RELATION TO OTHER WORK

As can be seen in Figure 1C, the effective receptive field can cover the entire image. However, the actual receptive field depends on the parameters of the RNNs, and can be learnt adaptively. As an insight to what is learnt, consider a separable filter, such as an axis aligned 2D Gaussian. Such filters can be applied exactly by a composition of 1D Gaussian convolutions in the horizontal and vertical directions. The 1D spatial RNNs can approximate finite 1D convolutions of this type.

We next discuss the relation of the L-RNN to prior work. First, ReNets (Visin et al., 2015), which is an architecture completely made of 1D RNNs (i.e. no CNNs). In ReNets, the input images are first split into non-overlapping patches of size $m \times n \times d$, where $m, n, d$ refer to width, height and feature channels respectively. The 1D RNNs takes the flattened patch ($mn \times d$) as input, and outputs feature vector of size $D \times 1$, where $D$ refers to the number of nodes used in the RNNs. In contrast, we interleave the L-RNN and CNN modules. There are two benefits of this: first, CNNs are more efficient at capturing local features than RNNs, the L-RNN stacked upon them is able to learn dependencies between local features (rather than the input channel reformatted); second, we are able to introduce more non-linearities *between* the hierarchical layers (through the convolutional+ReLU and pooling layers), and a RNN provides non-linearities *within* the same layer.

The 2D-RNN, proposed in (Graves & Schmidhuber, 2009; Theis & Bethge, 2015), is able to scan across the image or feature maps row-by-row, or column-by-column sequentially, with each RNN node accept input from three sources, namely, projections of current input, and feedbacks from the two neighbour nodes. By contrast, we use unidirectional 1D spatial RNNs, with each hidden node only accepting feedbacks from its previous node. Another advantage of our model is that rows or columns can be processed in parallel on GPUs, and training time is shortened.

Bell et al. (2016) (Inside-Outside Net) and Visin et al. (2016) (ReSeg) describe similar ideas for object detection and semantic segmentation. Both architectures follow a pipeline that consists of a CNN feature extractor (VGG Net) followed by spatial RNNs at the final prediction stage. In contrast, we treat the L-RNN module as a general computational layer, that can be inserted into any layer of modern architectures, and interleaved with CNN modules. This enables a network to be capable of learning contextual information in a flexible way at multiple levels, rather than with hand-crafted kernel sizes and receptive fields.

Note that the vanilla RNN unit consists of two terms, a local term and a recurrence term, where the local term is exactly the convolution operation. Therefore, the spatial RNN can be seen as a generalisation of the convolutional layer, and in the worst case, when the RNN learns no context, the layer simply becomes a convolutional one. For tasks with limited data (semantic segmentation in our case), we propose a regime for inserting the L-RNN into the pre-trained FCN and fine-tuning the entire network end-to-end. This means that we directly increase the representational power of the model, and set the pre-trained model free to learn contextual information if it is needed.

## 3 CNNS & LAYER-RNN MODULES

In this section, we describe the architecture for incorporating 1D spatial RNNs into the computational block of a Residual Networks(He et al., 2016b), and also discuss fusion methods for such blocks.

We start with the standard residual block of He et al. (2016b) (Figure 2(a)), and then replace the included CNN layer with bidirectional spatial RNNs, to includ a L-RNN module instead.

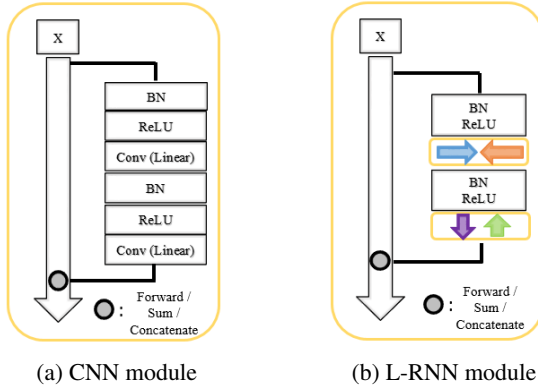

(a) CNN module
(b) L-RNN module

Figure 2: **Basic Modules for Classification:** CNN module is defined in a same way as ResNet (He et al., 2016b). L-RNN module is defined as a cascade of bidirectional spatial RNNs.
*Forward*, *Sum* or *Concatenation* can be used for skip layers. Batch Normalizations are used when training the architecture from scratch (Section 5.1).

We consider three fusion options for combining the features from such blocks with the input to subsequent layers; namely forward, sum and concatenation. Forward refers to the traditional feed-forward architectures:

$$X^{L+1} = F(X^L, W) \qquad (2)$$

i.e. the block simply becomes a new layer; sum denotes the method of the original residual networks:

$$X^{L+1} = X^L + F(X^L, W) \qquad (3)$$

so that the L-RNN module acts as a residual block; whilst, in concatenation, features from multiple layers (same spatial sizes) are concatenated:

$$X^{L+1} = [X^L; F(X^L, W)] \quad \textbf{(;)} \text{ refers to concatenation} \qquad (4)$$

Therefore, the channels of output feature maps will be the sum of the channels of the two concatenated layers (the number of parameters will be increased for the next layers). In the experimental evaluation of Section 5.1 we compare these options.

## 4 ADDING A LAYER-RNN TO A PRE-TRAINED CNN

In this section, we describe how a Layer-RNN *module*, can be seamlessly inserted into a pre-trained CNN. In a typical scenario, the CNN would be trained for classification on ImageNet (where there are copious annotations). After inserting the L-RNN modules, the hybrid L-RNN *network* can then be fine tuned for a new task such as pixel-level prediction, e.g. semantic segmentation (where the annotated data is usually more limited). This trick naturally allows multi-level contextual information to be effortlessly incorporated. Avoiding training the network from scratch means the entire network can be re-purposed with the available annotations and trained end-to-end for the new task, whilst benefiting from the earlier classification training.

We illustrate the idea using 1D convolution, but the same principles hold for the entire L-RNN module. As shown in Figure 3, the canonical CNN architecture for a 1D convolution can be denoted as:

$$X^{L+1} = f(W * X^L + b) \qquad (5)$$

where $*$ refers to convolution, $W$ and $b$ are the parameters of the CNN, $L, L+1$ denote the layer. The 1D spatial RNN can be written as :

$$X_i^{L+1} = f(U * X_i^L + V X_{i-1}^{L+1} + b) \qquad (6)$$

where $U, V, b$ refer to the parameters that are shared across the whole scan-line.
Notice that the 1D spatial RNN are designed to incorporate two terms, projections from local region (input-to-hidden) and recurrence term from previous hidden unit (hidden-to-hidden). In fact, it is

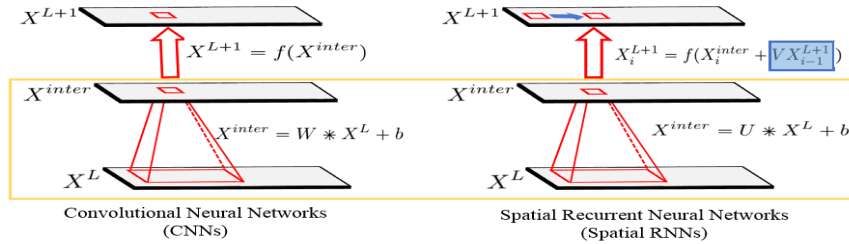

Figure 3: **CNNs & Spatial RNNs**
Spatial RNNs can be re-expressed as a two-step process, CNNs (Local features) + Recurrence.
The similarity between CNNs and spatial RNNs is highlighted by the yellow box.
The difference between CNNs and spatial RNNs is shown in blue box and arrow.

the presence of non-zero recurrence matrix $V$, that characterizes the 1D spatial RNN, and they can
be calculated in a two-step way as :

$$X^{inter} = U * X^L + b \quad (\text{ Convolution }) \tag{7}$$

$$X_i^{L+1} = f(X_i^{inter}) \quad (\text{ i = 1, zero initial states }) \tag{8}$$

$$X_i^{L+1} = f(X_i^{inter} + V X_{i-1}^{L+1}) \quad (i > 1) \tag{9}$$

By interpreting the recurrence in this way, 1D spatial RNNs can be constructed by inserting recurrence directly into any convolutional layer right after the convolution. If the recurrence matrix $V$ is initialized as zero, and ReLU is the activation function, then the 1D spatial RNN will be initialized exactly as the pre-trained CNNs. The complete L-RNN can be constructed by inserting two bidirectional spatial RNNs into subsequent layers of the pre-trained CNNs. We derive the expression of the *within-layer* gradient for use in back-prop fine-tuning in Appendix B.

## 5 EXPERIMENTAL EVALUATION

We test the proposed Layer-RNN on two supervised learning tasks: CIFAR-10 classification in Section 5.1; and PASCAL VOC 2012 segmentation in Section 5.2.

### 5.1 IMAGE CLASSIFICATION

In this section, we investigate classification performance under variations in an architecture containing L-RNN modules. We vary the depth of the network, the number and position of the L-RNN modules, the type of recurrent units in RNNs, the pooling mechanisms for the last pooling layer, and the method of fusing the block outputs.

**Architectures:** An overview of the architectures is given in Table 1 (with the fundamental building modules (CNN and L-RNN) adapted from Figure 2).

There are two principal architectural variations. The first variation is that from Network A to D, we gradually increase the network depth by adding CNN Modules, with the L-RNN module always stacked at the final stage to capture global information over the entire image, in a similar manner to the fully connected layers or average pooling in other networks. Network A has 5 convolutional layers.

The second principal variation, in Network E and F, is to interleave CNN and L-RNN modules. This means that the network is capable of learning representations across large spatial footprints at any stage in the network. To show the effectiveness of adding L-RNN modules, we include a Baseline-CNN composed of only convolutional layers (7 layers, with concatenation used at every skip layer). Network E is built upon the Baseline-CNN by inserting L-RNN modules before CNN modules at multiple stages. To make sure the performance gain is not from the increased number of parameters, we cut down the number of filters in the last CNN module to 128 (this number is 256 in the Baseline-CNN). Network F, uses more convolutional layers interleaved with L-RNN modules.

| Baseline-CNN | A | B | C | D | E | F |
|---|---|---|---|---|---|---|
| input (32 × 32 × 3) | | | | | | |
| convolution (3 × 3 × 64) | | | | | | |
| CNN Module (3 × 3 × 64) Concatenate | CNN Module (3 × 3 × 64) Feature Fusion | CNN Module (3 × 3 × 64) Forward CNN Module (3 × 3 × 64) Concatenate | CNN Module (3 × 3 × 64) Forward CNN Module (3 × 3 × 64) Forward CNN Module (3 × 3 × 64) Concatenate | CNN Module (3 × 3 × 64) Forward CNN Module (3 × 3 × 64) Forward CNN Module (3 × 3 × 64) Concatenate CNN Module (3 × 3 × 128) Forward CNN Module (3 × 3 × 128) Forward CNN Module (3 × 3 × 128) Concatenate | CNN Module (3 × 3 × 64) Concatenate | CNN Module (3 × 3 × 64) Concatenate CNN Module (3 × 3 × 64) Concatenate CNN Module (3 × 3 × 64) Concatenate |
| MaxPooling (2) | | | | | | |
| CNN Module (3 × 3 × 128) Concatenate | CNN Module (3 × 3 × 128) Feature Fusion | CNN Module (3 × 3 × 128) Forward CNN Module (3 × 3 × 128) Concatenate | CNN Module (3 × 3 × 128) Forward CNN Module (3 × 3 × 128) Forward CNN Module (3 × 3 × 128) Concatenate | CNN Module (3 × 3 × 128) Forward CNN Module (3 × 3 × 128) Forward CNN Module (3 × 3 × 128) Concatenate | LRNN Module (128) Forward CNN Module (3 × 3 × 128) Concatenate | LRNN Module (128) Forward CNN Module (3 × 3 × 64) Concatenate LRNN Module (128) Forward CNN Module (3 × 3 × 64) Concatenate |
| MaxPooling (2) | | | | | | |
| CNN Module (3 × 3 × 256) Concatenate | LRNN Module (256) Feature Fusion | LRNN Module (256) Concatenate | LRNN Module (256) Concatenate | LRNN Module (256) Concatenate | LRNN Module (128) Forward CNN Module (3 × 3 × 128) Concatenate | LRNN Module (128) Forward CNN Module (3 × 3 × 64) Concatenate LRNN Module (128) Forward CNN Module (3 × 3 × 64) Concatenate |
| Global Pooling (8) | | | | | | |
| Dropout (0.5) | | | | | | |
| Softmax (10) | | | | | | |

Table 1: **Network architectures for CIFAR-10 experiments**
In *Network A*, a variety of selections are tested (coded as blue). In Feature Fusion, we may choose Forward, Sum, Concatenation; in the LRNN module, GRU and vanilla RNNs are tested; max pooling or average pooling can be used for global pooling.
From *Network A* to *D*, the depth of networks is gradually increased by adding CNN modules, for example, comparing C to B, two more CNN modules are added based on B (coded as red). Comparing *Networks E and F* with the the Baseline-CNN, LRNN modules (green) are interleaved with CNN modules.

Other variations of architectures include: firstly, we may use Forward, Sum, Concatenation to fuse features; secondly, GRU and vanilla RNN units are compared for the L-RNN modules, ReLU is used for both cases as the non-linear activation; thirdly, both max pooling and average pooling are tested as global pooling. For clarity, we name the networks by these variations in Table 2. When Forward is selected to fuse features, Network A-Forward simply follows the traditional CNN with pure feed-forward layers. A-Concat uses concatenation as an alternative, and A-Sum follows the idea of residual networks proposed in (He et al., 2016b), the number of filters is gradually increased as the networks get deeper. To match dimensions for summation, $1 \times 1$ convolution is used in A-Sum. In our experiments, we found that concatenation works better than sum (Table 2). Therefore, in all

other architectures (B,C,D), as we gradually increase the network depth by adding CNN modules, we fuse the skip layers by only alternating between concatenation and forward.

Following the VGG-net (Simonyan & Zisserman, 2015), in all architectures, convolutional kernels in the CNN Module are of size $3 \times 3$. Maxpoolings ($2 \times 2$) are used as intermediate pooling, and $8 \times 8$ global poolings (average or max) are applied at the end. To avoid overfitting, we use dropout (0.5). Training details and recurrent units are described in the Appendix A. Implementations are mostly based in Theano (Theano Development Team, 2016) with single NVIDIA Titan X.

**Dataset & Evaluation.**   We conducted experiments on the CIFAR-10 dataset, which consists of 40k training images, 10k validation and 10k testing images in 10 classes, and each of the image is of $32 \times 32$ pixels with RGB channels. We augment the training data with simple transformations (rotation, flipping, scaling) on the fly. The mean image over the whole training set is subtracted from each image during training. Following the standard evaluation protocol, we report the *top1* error on the testing set.

**Results & Discussion.**   We present detailed comparisons with other published methods in Table 2.

| CIFAR-10 | # Params | # Conv Layers | Approx. Time / Epoch (s) | Top1 Error(%) |
|---|---|---|---|---|
| ReNet (Visin et al., 2015) | – | 0 | – | 12.35 |
| NIN (Lin et al., 2013) | – | – | – | 8.81 |
| FitNet (Romero et al., 2014) | 2.5M | 19 | – | 8.39 |
| Highway (Srivastava et al., 2015) | 2.3M | 19 | – | 7.54 |
| ResNet-110 (He et al., 2016a) | 1.7M | 110 | – | 6.61 |
| ResNet-164 (He et al., 2016b) | 1.7M | 164 | – | 5.46 |
| Dense Net (Huang et al., 2016) | 27.2M | 100 | – | **3.74** |
| | | | | |
| **Baseline-CNN-Avg** | 1.56M | 7 | 331 | 9.07 |
| **Baseline-CNN-Max** | 1.56M | 7 | 331 | 8.48 |
| **A-Concat-RNN-Avg** | 0.9M | 5 | 293 | 7.65 |
| **A-Concat-RNN-Max** | 0.9M | 5 | 293 | 7.43 |
| **A-Forward-GRU-Max** | 1.68M | 5 | 315 | 7.57 |
| **A-Concat-GRU-Max** | 1.95M | 5 | 377 | 7.35 |
| **A-Sum-GRU-Max** | 1.99M | 5 | 383 | 7.69 |
| **B-GRU-Max** | 2.3M | 9 | 542 | 6.62 |
| **B-RNN-Max** | 1.27M | 9 | 483 | 6.78 |
| **C (GRU-Max)** | 2.5M | 13 | 726 | 6.21 |
| **D (GRU-Max)** | 3M | 19 | 1321 | 5.73 |
| **E (RNN-Max)** | 0.97M | 7 | 462 | 5.96 |
| **F (RNN-Max)** | 1.55M | 15 | 394 (Tensorflow on 2 GPUs) | **5.39** |

Table 2: **Comparison with previous published methods on CIFAR-10**
The networks are named by the chosen operation at every step; for instance, *A-Forward-GRU-Max*, refers to the architecture A with Forward feature fusion, GRU in L-RNN Module, and max pooling as the final global pooling.

From the experimental results, we can draw the following conclusions:

**Comparison of basic choices.**   Max pooling consistently performs better when used as the global pooling in our case, this is seen in the results by Baseline-CNN-Avg (9.07%) vs. Baseline-CNN-Max (8.48%), and A-Concat-RNN-Avg (7.65%) vs. A-Concat-RNN-Max (7.43%). One possible explanation would be that for classification tasks, decisions are based on the most salient features.

In our experiments for shallow networks, the summing of residual connections shows no benefit compared to feed-forward or concatenation. This observation is made from the results by A-Forward-GRU-Max (7.57%), A-Concat-GRU-Max (7.35%) and A-Sum-GRU-Max (7.69%). Thus, as also employed in U-Net or DenseNet (Ronneberger et al., 2015; Huang et al., 2016), concatenation can be used as an alternative to summation in building deeper networks.

It can be seen that vanilla RNN units trained with Layer Normalization (Ba et al., 2016) can perform almost as well as GRU, while saving a a large number of parameters (by comparing the results from A-Concat-RNN-Max with $0.9M$ parameters ($7.43\%$) and that of A-Concat-GRU-Max with $1.95M$ parameters ($7.36\%$), B-RNN-Max with $1.27M$ parameters ($6.78\%$) vs. B-GRU-Max with $2.3M$ parameters ($6.62\%$)).

**Networks with L-RNN module stacked at the final stage.** Even shallow networks with L-RNN modules (architectures A) can achieve comparable or superior performance to deep architectures with 19 layers that requires more parameters (e.g. Network A-Concat-RNN-Max ($0.9M$) vs. Highway($2.3M$)). This confirms that when a L-RNN module is stacked on top of CNNs, it is able to capture global information, avoiding the multiple layer route to increasing receptive fields in standard architectures, e.g. in (Romero et al., 2014; Srivastava et al., 2015).

As expected, networks can always improve classification performance by adding more CNN modules (going from architecture A to D). Network D with 19 convolutional layers performs better than the ResNet-110 (by $0.3\%$ *top1* error), (though Network D has more parameters than the ResNet-110) and is slightly worse than ResNet-164 (by $0.25\%$ *top1* error). Thus, following this trend, it is reasonable to expect a benefit if L-RNN Modules are combined with very deep networks, like the residual variants.

**Networks with L-RNN modules interleaved with CNN modules.** Comparing the performance of Baseline-CNN-Max ($8.48\%$) with that of Network E ($5.96\%$), there is a significant performance boost ($2.5\%$), brought by simply inserting L-RNN modules. Network E also has other advantages over the networks A to D: the number of parameters, network depth, and running time. Furthermore, when we continue increasing the network depth and interleaving L-RNN modules, **Network F achieves comparable results** ($5.39\%$) **to ResNet-164** ($5.46\%$) **and with fewer parameters** ($1.55M$ **vs.** $1.7M$). This confirms that, firstly, L-RNN modules can be combined with very deep networks; and secondly, rather than hand-craft the kernel size, we should set the model free and learn contextual information at any stage.

## 5.2 Semantic Segmentation

In this section, we insert L-RNN modules into the VGG-16 networks (pre-trained on ImageNet (Deng et al., 2009)), and fine-tune the entire network for the PASCAL VOC 2012 segmentation task. The objective is to boost the segmentation performance by providing contextual information via the L-RNNs. In particular, we consider the two FCN segmentation architectures originally introduced by Long et al. (2015), FCN-32s and FCN-8s; these are described below.

We proceed in three steps: first, we establish baselines by training our own FCN-32s and FCN-8s (Appendix C), and comparing their performance to those of (Long et al., 2015). We also investigate the loss in performance as the fully connected (FC) layer is gradually reduced from 4096 to 512 channels. The reason for doing this is that when we insert the L-RNN module, its complexity (dimension of the hidden units) depends on this number of channels, and so the overall complexity can be varied. In the second step, we insert L-RNNs into the FCN-32s architecture and evaluate the change in performance. Finally, we insert L-RNNs into the FCN-8s architecture and compare with previous published methods.

**Dataset & Evaluation.** We used a training set consisted of VOC2012 training data (1464 images provided by the challenge organizers), and augmented with training and validation data from Hariharan et al. (2014), which further extend the training set to a total of $11,685$ images with pixel-level annotation. After removing the overlapping images between VOC2012 validation data and this dataset, we are left with $346$ images from the original VOC2012 validation set to validate our model. In all the following experiments, we use a single scale for the input images ($384 \times 384$), and only horizontal flipping is used for data augmentation. The performance is measured in terms of pixel intersection-over-union (IOU) averaged across the 21 classes.

### 5.2.1 BASELINE ARCHITECTURES AND TRAINING

**Architecture & Training.** In the FCN-32s, input images are passed through the whole networks, and end up with predictions of $12 \times 12 \times 21$, then, up-sampling layers are directly used to map the predictions back to $384 \times 384$ (32 times). In the FCN-16s, instead of directly up-sampling 32 times, the predictions are first up-sampled by 2, and summed up with stream predictions from pool4 (named after VGG16), then up-sampled by 16 times. In the FCN-8s, the stream predictions from pool3 are further added to the results from FCN-16s, thus, up-sampling layers with only factor 8 is needed. (Appendix C)

For all the architectures, the base net(VGG16) is pre-trained on ImageNet (Deng et al., 2009), we further train on Pascal VOC2012 for 50 epochs, similar to the experiment for CIFAR-10, we iteratively increase or decrease the learning rate between $10^{-3}$ and $10^{-5}$ after every 10 epochs. The 4096 channel architectures are trained first, and then the number of channels is gradually reduced in the FC layer by randomly cutting them (e.g. from 4096 to 2048), and re-training the networks.

**Results & Discussion.** Table 3 shows the performance of the six baselines: FCN-32s and FCN-8s with the number of channels varying from 512 to 4096. We observe that reducing the nodes in the FC layers does produce a performance drop (from 4096 to 1024 nodes, $1\%$ mean IOU) in both FCN-32s and FCN-8s. Although from $1024$ to $4096$ nodes, the improvement is tiny, the difference in the number of parameters is over 64 million. Consequently, in the following experiments we choose to perform experiments based on networks with $512, 1024$ or $2048$ channels only (i.e. not 4096). In comparison to the original performance for the FCN-8s architecture in (Long et al., 2015), we exceed this (by 64.4 to 61.3 mean IOU) in our training. Thus, we use our trained networks as a baseline.

### 5.2.2 FCN-32S WITH L-RNN MODULES

**Architecture & Training.** The architecture FCN-32s(L-RNN) is shown in figure 4, the convolutional part of the architecture is initialized with the pre-trained FCN-32s(2048 channels in FC layer) baseline. Then, two 1D spatial RNNs are inserted into the fc1 layer in the horizontal direction, and two 1D spatial RNNs are inserted into the fc2 layer in the vertical direction. The convolution activations of fc1 are shared for both left-right and right-left scanning. Similarly for fc2, the convolution activations are shared for up-down and down-up scanning. Thus the fc1 and fc2 layers together with the added 1D spatial RNNs form a complete L-RNN module.

During training, as described in section 4, the 1D spatial RNNs are initialized with a zero recurrence matrix. The entire network is then fine-tuned end-to-end with the PASCAL VOC2012 data. We adopt RMS-prop (Tieleman & Hinton, 2012) for 30 epochs with hyper-parameters $lr = 10^{-4}$, $\rho = 0.9, \epsilon = 10^{-8}$, then decrease the learning rate to $lr = 10^{-5}$ for 10 epochs.

**Results & Discussion.** The results are shown in Table 3. Compare the 32s rows with and without the L-RNN for the FC layers with 512, 1024, and 2048 channels. As can be seen, the addition of the L-RNN always improve the segmentation performance over the pre-trained FCN-32s baselines. However, the improvement is not large – about $1 - 1.5\%$ mean IOU. This is because the receptive field in the fully connected layers of FCN-32s is sufficiently large to cover $224 \times 224$ pixels of the input patch, and consequently the networks are not able to benefit much from the context provided by the L-RNN. The benefit is greater when L-RNNs are added to the lower layers (where the receptive fields of the convolutions is much smaller), and we turn to that case next.

### 5.2.3 FCN-8S WITH L-RNN MODULES

**Architecture & Training.** The architecture FCN-8s(L-RNN) is shown in figure 4, as with the FCN-32s architecture, 1D spatial RNNs are inserted into the fc1 and fc2 layers to form a L-RNN module. L-RNNs are also inserted into the lower layers, namely pool3 and pool4 layers. Unlike the FC layers in the FCN-32s, where prediction for each central pixel comes from image patches of size $224 \times 224$, the predictions from pool3 and pool4 are based on receptive field on the image of much smaller sizes (around $44 \times 44$ and $100 \times 100$ pixels respectively). Thus, the inserted L-RNN modules must be able to model relatively long-range dependencies.

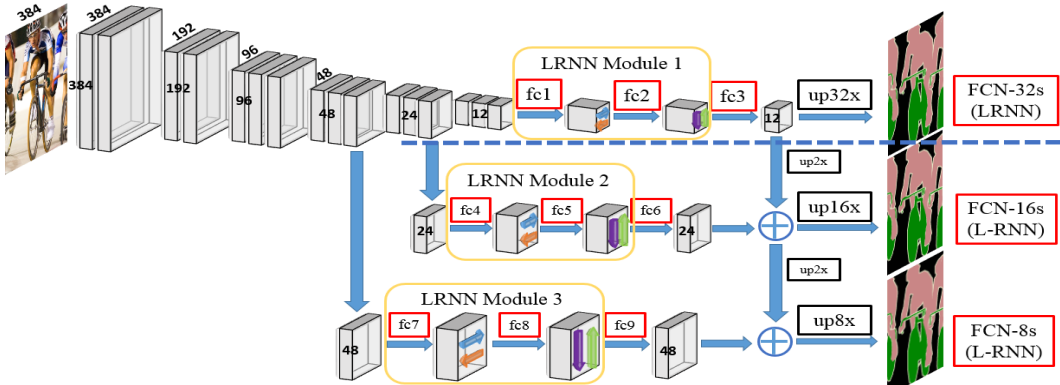

Figure 4: FCN-32s (above the blue dash line) and FCN-8s with L-RNN modules.
Spatial RNNs are inserted to the fully connected (FC) layers in all FCNs, every two FC layers construct a complete L-RNN module.
$\{384, 192, 96\}$ indicate the spatial sizes of the feature maps.
Kernel Sizes for the fully connected layers (n is an experimental variable– number of channels) :
fc1 : $7 \times 7 \times 512 \times \quad n$ , fc2 : $1 \times 1 \times \quad n \quad \times \quad n$ , fc3 : $1 \times 1 \times \quad n \quad \times 21$
fc4 : $1 \times 1 \times 512 \times 1024$, fc5 : $1 \times 1 \times 1024 \times 1024$, fc6 : $1 \times 1 \times 1024 \times 21$
fc7 : $1 \times 1 \times 256 \times 1024$, fc8 : $1 \times 1 \times 1024 \times 1024$, fc9 : $1 \times 1 \times 1024 \times 21$

During training, the network is initialized from the FCN-8s baseline, and then fine-tuned using segmentation data. Again the PASCAL VOC dataset is used. Furthermore, when comparing to the other previously published methods, the network is further trained on the COCO trainval dataset, and we use a densely connected CRF as post-processing (Krhenbhl & Koltun, 2012).

**Results on PASCAL VOC Validation set.** The experimental results are shown in Table 3.

| Type | # of channels in FC | L-RNNs added | Pixel Acc % | Mean IOU % |
|---|---|---|---|---|
| 32s | 512 | NO | 90.4 | 61.5 |
| 32s | 1024 | NO | 90.5 | 62.1 |
| 32s | 2048 | NO | 90.7 | 62.7 |
| 32s | 4096 | NO | 90.7 | 62.9 |
| 8s | 1024 | NO | 91.3 | 63.8 |
| 8s | 2048 | NO | 91.2 | 64.1 |
| 8s | 4096 | NO | 91.3 | 64.4 |
| 8s (original (Long et al., 2015)) | 4096 | – | – | 61.3 |
| 32s | 512 | YES | 90.8 | 62.7 |
| 32s | 1024 | YES | 90.9 | 63.4 |
| 32s | 2048 | YES | 91.1 | 64.2 |
| **8s** | **2048** | **YES** | **92.6** | **69.1** |

Table 3: Comparison of FCN networks on the PASCAL VOC2012 segmentation **validation** set.

Comparing the rows for 32s with and without L-RNN, to those for 8s with and without L-RNN. We can draw the following conclusions:

**Improvement due to the skip layers.** It can be seen (for IOU) that going from FCN-32s(2048) to FCN-8s(2048), where there are additional skip layers, the performance is boosted from 62.7 to 64.1. The skip layers in the FCN-8s architecture introduce more parameters, but this is not the reason for the performance boost since FCN-8s(2048) and FCN-32s(4096), have a similar number of parameters though they perform very differently (64.1 vs. 62.9). This observation confirms that the performance gain is brought by the the skip layers, rather than the increased number of parameters.

**Improvement due to L-RNN module.** Inserting a L-RNN to the FC layers of FCN-32s(2048), only improves the performance from 62.7 to 64.2. However, as noted earlier, since the nodes in the

FC layers already cover the entire input patch of size $224 \times 224$, the L-RNN can contribute only little context here.

In contrast, adding L-RNNs to FCN-8s brings a substantial improvement from $64.1$(FCN-8s) to $69.1$(FCN-8s-LRNN). This process will introduce more parameters due to the recurrence term in the RNNs, but it is clear that the improvement is mainly from the inserted L-RNN module after pool3 and pool4 in FCN-8s, rather than from the increased number of parameters. The reason is that, when comparing FCN-8s (2048 channels without L-RNN) to FCN-8s (4096 channels without L-RNN), although the number of parameters is increased dramatically, the performance is only increased from $64.1$ to $64.4$. While FCN-8s (4096 channels without L-RNN) has roughly the same number of parameters as that of FCN-8s (2048 channels with L-RNN), but the performance gain is from $64.4$ to $69.1$. In conclusion, the L-RNN is able to learn contextual information over a much larger range than the receptive field of pure local convolutions.

**Results on PASCAL VOC Test set.** Table 4 shows the results of the FCN-8s with L-RNNs on the PASCAL VOC test data, and also compares to others who have published on this dataset. The performance is far superior to the original result (Long et al., 2015) using a FCN-8s with 4096 channels (whereas only 2048 channels are used here). We also compare to the dilated convolution network of (Yu & Koltun, 2016), obtaining comparable, though slightly better performance. Note that in (Yu & Koltun, 2016), multi-scale contextual information is captured by explicitly designing dilated convolution kernels, while the L-RNN is able to learn contextual information implicitly. Finally, we compare to (Zheng et al., 2015) who add a densely connected CRF to FCN-8s. If we also add a dense CRF as post-processing, we boost the performance by $1\%$ in IOU (the same boost as obtained by (Yu & Koltun, 2016)). In Figure 5, we show the samples of semantic segmentations on

| Methods | Mean IOU % | | | |
| --- | --- | --- | --- | --- |
|  | **P** | **P+CRF** | **P+COCO** | **P+COCO+CRF** |
| FCN-8s (Long et al., 2015) | 62.2 | n/a | n/a | n/a |
| CRF-RNNs (Zheng et al., 2015) | n/a | 72.0 | n/a | 74.7 |
| Dilated Conv. (Yu & Koltun, 2016) | n/a | n/a | 73.5 | 74.7 |
| FCN-8s-LRNN (2048) | **71.9** | **72.7** | **74.2** | **75.7** |

Table 4: Comparison of mean IOU on the PASCAL VOC2012 segmentation **Test** set.
(All results are based on VGG-16 net)
Training is on P: PASCAL VOC2012;       COCO: COCO dataset.
`http://host.robots.ox.ac.uk:8080/anonymous/YJBLI7.html`

the PASCAL VOC2012 validation set. In each figure, we show our predictions and the results after CRF post-processing. Comparing with the end-to-end trainable CRF-RNN (Zheng et al., 2015), our predictions miss the small details, like the wheel of the bicycle, but show much better performance in determining the class of the segmented regions – something that context can really contribute to.

# 6  CONCLUSION & FUTURE WORK

This paper has shown that the proposed L-RNN module is an alternative way of adding multi-level spatial context to a network. In fact, L-RNNs can be interleaved with convolutional layers to learn context at any stage. When the L-RNN is only used at the final stage after the CNNs, it gives shallow networks the receptive fields of far deeper networks. Furthermore, we have demonstrated that inserting L-RNNs can boost the performance of pre-trained networks, and given an initialization procedure that makes this training a simple matter of end-to-end fine tuning.

There is much left to investigate using L-RNNs as a new building block, and we suggest some avenues here: (i) training the hybrid architectures on larger dataset, such as ImageNet (Deng et al., 2009), and learn representations that can be transferred to other vision tasks, (ii) a similar investigation for deep residual networks where the residual blocks are either convolutional or L-RNNs; and (iii) including a CRF final layer in end-to-end training.

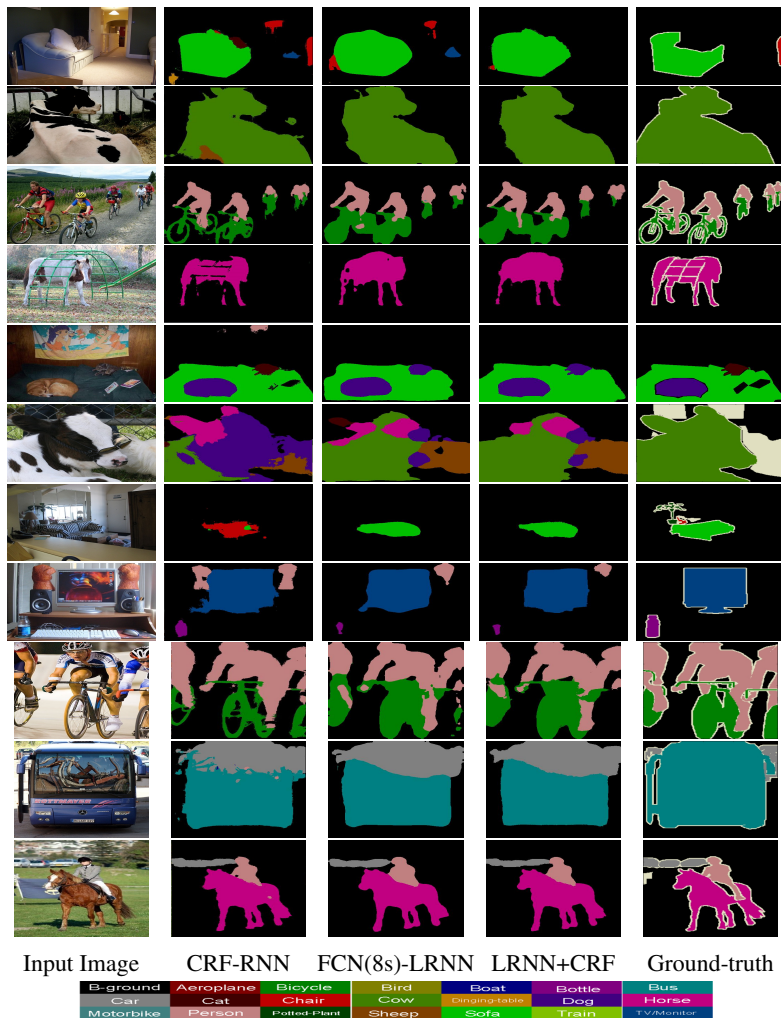

Input Image CRF-RNN FCN(8s)-LRNN LRNN+CRF Ground-truth

Figure 5: **Qualitative Results.** First column: input image. Second column: prediction from Zheng et al. (2015). Third column: prediction from the our networks. Fourth column: CRF post-processing. Fifth column: ground-truth annotation.

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

# Appendices

## A   TRAINING DETAILS FOR CIFAR-10

### A.1   RNN WITH GATED RECURRENT UNITS

In the Layer-RNN, we test the gated recurrent units (GRU) for the RNN blocks (Chung et al., 2015), the GRU has two gates, namely reset gate $r$ and update gate $z$. Intuitively, the reset gate determines how to combine the new input with the previous memory, and the update gate defines how much of the previous memory to use, thus, the hidden state $s_t$ of the GRU at time $t$ can be computed as :

$$z = \sigma(x_t U^z + s_{t-1} W^z) \tag{10}$$
$$r = \sigma(x_t U^r + s_{t-1} W^r) \tag{11}$$
$$h = f(x_t U^h + (s_{t-1} \circ r) W^h) \tag{12}$$
$$s_t = (1 - z) \circ h + z \circ s_{t-1} \tag{13}$$

### A.2   VANILLA RNN WITH LAYER NORMALIZATION

To simplify the training process and reduce number of parameters, we also test the vanilla RNNs for the RNN blocks with Layer Normalization(Ba et al., 2016). In a standard RNN, the outputs in the recurrent layer are calculated from the current input $x_t$ and the previous hidden states $h_{t-1}$, which are denoted as $a_t = U x_t + V h^{t-1}$. The layer normalized layer is computed as :

$$h_t = f(\frac{g}{\sigma_t} \circ (a_t - \mu_t) + b) \tag{14}$$

$$\mu_t = \frac{1}{H} \sum_{i=1}^{H} a_t^i \qquad \sigma_t = \sqrt{\frac{1}{H} \sum_{i=1}^{H} (a_t^i - \mu_t)^2} \tag{15}$$

Where $U$ is the current input-to-hidden term, and $V$ is the hidden-to-hidden recurrence term, $b$ and $g$ are defined as the bias and gain parameters of the same dimension as $h_t$.

During training, we iteratively increase and decrease the learning rate (learning rate restart) between $10^{-3}$ and $10^{-5}$ based on the conjecture that (Figure 6), networks tend to get trapped in the regions with small derivatives, such as saddle points or bad local minima Dauphin et al. (2014). Traditionally, the learning rate is decreased every several epochs, and gradients that are used to update parameters depend on both the learning rate and the derivatives *w.r.t* loss functions. At the end of training, both of these two terms tend to be very small. Therefore, it becomes difficult for the networks to escape from these regions. During our training, we restart the learning rate every some epochs (we try 60 or 80 in our training), and decrease it gradually.

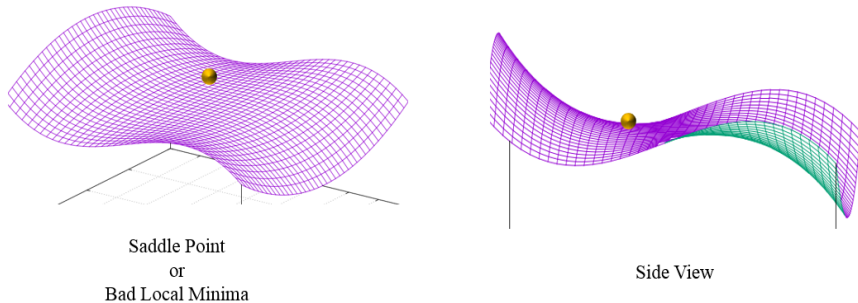

Saddle Point
or
Bad Local Minima

Side View

Figure 6: Intuitive Loss Surfaces.
Deep Neural Networks may easily be trapped into saddle point or bad local minima.

## B  FINE-TUNING LAYER-RNNS WITH ZERO RECURRENCE MATRIX

In this section, we derive the procedure for fine-tuning the recurrence matrix, when it is initialized as zeros. We will only consider 1D scan-lines of the spatial RNN, and therefore simplify the derivation to a 1D sequence. Consider the fully connected layer for simplicity, $L, L + 1$ denote layer, $t$ refers to the index of input, $f$ refers to ReLU, $U, V$ refer to the input-hidden matrix and recurrence matrix respectively.

$$s_t = U X_t^L + V X_{t-1}^{L+1} + b \tag{16}$$

$$X_t^{L+1} = f(s_t) \tag{17}$$

Assume $E$ denotes the loss function for a specific task. Since $V$ is shared for the whole 1D sequence (length denoted by T), the back-propagation *within* the layer $L + 1$ can then be derived as:

$$\frac{\partial E}{\partial V} = \sum_T \sum_{t \leqslant T} \frac{\partial E}{\partial X_T^{L+1}} \cdot \frac{\partial X_T^{L+1}}{\partial X_t^{L+1}} \cdot \frac{\partial X_t^{L+1}}{\partial s_t} \cdot \frac{\partial s_t}{\partial V} \tag{18}$$

where,

$$\frac{\partial X_T^{L+1}}{\partial X_t^{L+1}} = \frac{\partial X_T^{L+1}}{\partial X_{T-1}^{L+1}} \frac{\partial X_{T-1}^{L+1}}{\partial X_{T-2}^{L+1}} \cdots \frac{\partial X_{t+1}^{L+1}}{\partial X_t^{L+1}} \quad \text{and} \quad \frac{\partial X_{t+1}^{L+1}}{\partial X_t^{L+1}} = V^T \cdot diag(f') \tag{19}$$

Each Jacobian $\frac{\partial X_{t+1}^{L+1}}{\partial X_t^{L+1}}$ is a product of two matrices: (a) the recurrence weight matrix $V$, and (b) the diagonal matrix composed of the derivative of ReLU ($f'$). Therefore, when $V$ is initialized to zero at the starting point, there are no long-range dependencies, and for $t < T$, $\frac{\partial X_{t+1}^{L+1}}{\partial X_t^{L+1}}$ will be zero, while for $t = T$:

$$\frac{\partial E}{\partial V} = \sum_T \frac{\partial E}{\partial X_T^{L+1}} \cdot \frac{\partial X_T^{L+1}}{\partial s_T} \cdot \frac{\partial s_T}{\partial V} \quad \text{where} \quad \frac{\partial X_T^{L+1}}{\partial s_T} = f', \quad \frac{\partial s_T}{\partial V} = X_{T-1}^{L+1} \tag{20}$$

$$V_1 = V_0 - \alpha \frac{\partial E}{\partial V} \quad \text{gradient descent at first iteration} \tag{21}$$

Since $V_0$ is initialized as zero, $V_1 = -\alpha \frac{\partial E}{\partial V}$. In other words, instead of initializing the recurrence matrix $V$ randomly or to be identity matrix, we actually initialize it based on the features in a local neighbourhood (equation 20). During the back-propagation of spatial RNNs, gradients flow *within* layers, $\frac{\partial E}{\partial U}$ (*between layers*) is calculated in the same way as normal convolutional layers.

## C    DETAILED DESCRIPTION OF THE FCNS USED IN THE PAPER

The complete FCNs architecture used in the paper.

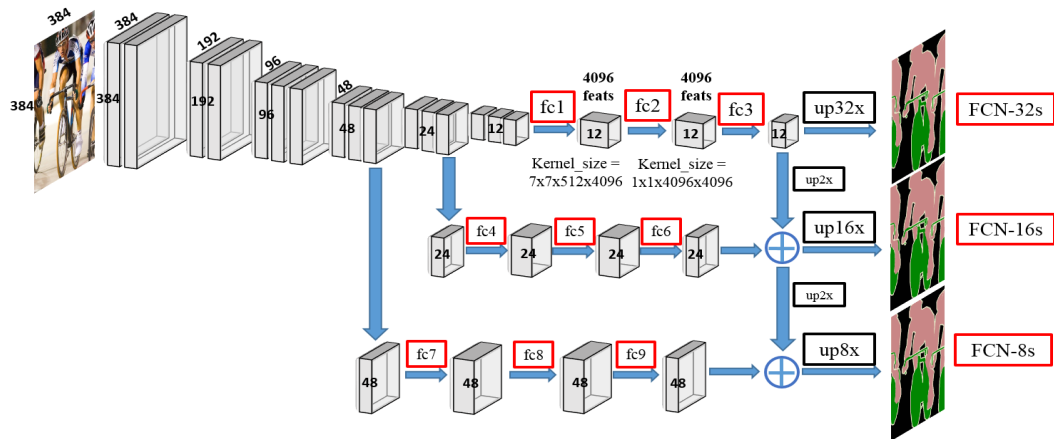

Figure 7: Complete FCNs used extensively in the paper
In FCN-32s, output feature maps of spatial size $12 \times 12$ is directly up-sampled by 32 times.
In FCN-16s, output feature maps of spatial size $12 \times 12$ is first up-sampled by 2, then sum up with the prediction scores calculated from feature maps of spatial size $24 \times 24$, and up-sample by 16 times.
In FCN-8s, the summed prediction scores are further up-sampled by 2, then sum up with the prediction scores calculated from feature maps of spatial size $48 \times 48$, and up-sample by 8 times.
Kernel Sizes for the fully connected layers :
fc1 : $7 \times 7 \times 512 \times 4096$,    fc2 : $1 \times 1 \times 4096 \times 4096$,    fc3 : $1 \times 1 \times 4096 \times 21$
fc4 : $1 \times 1 \times 512 \times 1024$,    fc5 : $1 \times 1 \times 1024 \times 1024$,    fc6 : $1 \times 1 \times 1024 \times 21$
fc7 : $1 \times 1 \times 256 \times 1024$,    fc8 : $1 \times 1 \times 1024 \times 1024$,    fc9 : $1 \times 1 \times 1024 \times 21$

