# Peer review of "Layer Recurrent Neural Networks"

_ICLR 2017 — rejected_

[Public Comment · (anonymous) · 20 Nov 2016]
**Addditional evaluation needed**

Hey,

I like the described approach to capture dependencies in images! 

Can you let me know if you did the following experiments:
* comparing both the performance and run-time of a single L-RNN layer compared with global average pooling
* comparing GRUs vs. LSTMs
* comparing a ResNet block with two recurrent layers as proposed with two dilated convolutional layers, which have recently been shown powerful for segmentation?
* evaluating L-RNN on other data sets than CIFAR-10, in particular using images of a higher resolution? The performance gain via L-RNN should be more pronounced for images larger than 32x32 at the cost of a higher runtime.

In your classification architectures, is it right that you are performing global max-pooling after the L-RNN layer? Did you try global average pooling or using the last hidden states of the recurrent layers?

I think the index i is missing in equation 6: (W * X^L)_i. I also suggest to polish the text by removing clutter (e.g. ‘as it is well known’, ‘where there are copious annotations’, ‘what is very clear’, ‘as practitioners know’, ‘it is worth noting’, ‘as expected’) and avoiding slang (e.g. can’t).

Best,

[Public Comment · (anonymous) · rating 7 · confidence 4 · 06 Dec 2016 (modified: 20 Jan 2017)]
**Additional evaluations required for being accepted**

The authors propose the use of a vertical and horizontal one-dimensional RNN (denoted as L-RNN module) to capture long-range dependencies and summarize convolutional feature maps. L-RNN modules are an alternative to deeper or wider networks, 2D RNNs, dilated (Atrous) convolutional layers, and a simple flatten or global pooling layer when applied to the last convolutional layer for classification. L-RNN modules are faster than 2D RNNs, since rows and columns can be processed in parallel, are easy to implemented, and can be inserted in existing convolutional networks. The authors demonstrate improvements for classification and semantic segmentation.

However, further evaluations are required that show for which use cases L-RNNs are superior to alternatives for summarizing convolutional feature maps:

1. I suggest to use a fixed CNN with as certain number of layers, and summarize the last feature map by a) a flatten layer, b) global average pooling, c) a 2D RNN, d) and dilated convolutional layers for segmentation. The authors should report both the run-time and number of parameters for these variants in addition to prediction performances. For segmentation, the number of dilated convolutional layers should be chosen such that the number of parameters is similar to a single L-RNN module.

2. The authors compare classification performances only on 32x32 CIFAR-10 images. For higher resolution images, the benefit of L-RNN modules to capture long-range dependencies might be more pronounced. I therefore suggest evaluating classification performances on one additional dataset with higher resolution images, e.g. ImageNet or the CUB bird dataset.

Additionally, I have the following minor comments:

3. The authors use vanilla RNNs. It might be worth investigating LSTMs or GRUs instead.

4. For classification, the authors summarize hidden states of the final vertical recurrent layer by global max pooling. Is this different from more common global average pooling or concatenating the final forward and backward recurrent states?

5. Table 3 is hard to understand since it mingles datasets (Pascal P and COCO C) and methods (CRF post-processing). I suggest, e.g., using an additional column with CRF ‘yes’ or ‘no’. I further suggest listing the number of parameters and runtime if possible.

6. Section 3 does not clearly describe in which order batch-normalization is applied in residual blocks. Figure 2 suggest that the newer BN-ReLU-Conv order described in He et al. (2016) is used. This should be mentioned in the text.

Finally, the text needs to be revised to reach publication level quality. Specifically, I have the following comments:

7. Equation (1) is the update of a vanilla RNN, which should be stated more clearly. I suggest to first describe (bidirectional) RNNs, to reference GRUs and LSTMs, and then describe how they are applied here to images. Figure 1 should also be referenced in the text.

8. In section 2.2, I suggest to describe Bell at al. more clearly. Why are they using eight instead of four RNNs? 

9. Section 4 starts with a verbose description about transfer learning, which can be compressed into a single reference or skipped entirely.

10. Equation (6) seems to be missing an index i.

11.In particular section 5 and 6 contain a lot of clutter and slang, which should be avoided:
11.1 page 8: ‘As can be seen’, ‘we turn to that case next’
11.2 page 9: ‘to the very high value’, ‘as noted earlier’,  ‘less context to contribute here’
11.3 page 10: ‘In fact’, ‘far deeper’, ‘a simple matter of’, ‘there is much left to investigate.

[Official Review · AnonReviewer2 · rating 6 · confidence 4 · 16 Dec 2016]
**No Title**

The paper proposes a method of integrating recurrent layers within larger, potentially pre-trained, convolutional networks. The objective is to combine the feature extraction abilities of CNNs with the ability of RNNs to gather global context information.
The authors validate their idea on two tasks, image classification (on CIFAR-10) and semantic segmentation (on PASCAL VOC12).

On the positive side, the paper is clear and well-written (apart from some occasional typos), the proposed idea is simple and could be adopted by other works, and can be deployed as a beneficial perturbation of existing systems, which is practically important if one wants to increase the performance of a system without retraining it from scratch. The evaluation is also systematic, providing a clear ablation study. 

On the negative side, the novelty of the work is relatively limited, while the validation is lacking a bit. 
Regarding novelty, the idea of combining a recurrent layer with a CNN, something practically very similar was proposed in Bell et al (2016). There are a few technical differences (e.g. cascading versus applying in parallel the recurrent layers), but in my understanding these are minor changes. The idea of initializing the recurrent network with the CNN is reasonable but is at the level of improving one wrong choice in the original work of Bell, rather than really proposing something novel. 
This contribution (" we use RNNs within layers") is repeatedly mentioned in the paper (including intro &  conclusion), but in my understanding was part of Bell et al, modulo minor changes. 

Regarding the evaluation, experiments on CIFAR are interesting, but only as proof of concept. 

Furthermore, as noted in my early question, Wide Residual Networks (Sergey Zagoruyko, Nikos Komodakis, BMVC16)
report  better results on CIFAR-10 (4% error), while not using any recurrent layers (rather using instead a wide, VGG-type, ResNet variant). So. 
The authors answer: "Wide Residual Networks use the depth of the network to spread the receptive field across the entire image (DenseNet (Huang et al., 2016) similarly uses depth). Thus there is no need for recurrence within layers to capture contextual information. In contrast, we show that a shallow CNN, where the receptive field would be limited, can capture contextual information within the whole image if a L-RNN is used."

So, we agree that WRN do not need recurrence - and can still do better. 
The point of my question has practically been whether using a recurrent layer is really necessary; I can understand the answer as being "yes, if you want to keep your network shallow".  I do not necessarily see why one would want to keep one's network shallow.

Probably an evaluation on imagenet would bring some more insight about the merit of this layer. 


Regarding semantic segmentation, one of my questions has been:
"Is the boost you are obtaining due to something special to the recurrent layer, or is simply because one is adding extra parameters on top of a pre-trained network? (I admit I may have missed some details of your experimental evaluation)"
The answer was:
"...For PASCAL segmentation, we add the L-RNN into a pre-trained network (this adds recurrence parameters), and again show that this boosts performance - more so than adding the same number of parameters as extra CNN layers - as it is able to model long-range dependences"
I could not find one such experiment in the paper ('more so than adding the same number of parameters as extra CNN layers'); I understand that you have 2048 x 2048 connections for the recurrence, it would be interesting to see what you get by spreading them over (non-recurrent) residual layers.
Clearly, this is not going to be my criterion for rejection/acceptance, since one can easily make it fail - but I was mostly asking for some sanity check 

Furthermore, it is a bit misleading to put in Table 3 FCN-8s and FCN8s-LRNN, since this gives the impression that the LRNN gives a  boost by 10%. In practice the "FCN8s" prefix of "FCN8s-LRNN" is that of the authors, and not of Long et al (as indicated in Table 2, 8s original is quite worse than 8s here). 

Another thing that is not clear to me is where the boost comes from in Table 2; the authors mention that "when inserting the L-RNN after pool 3 and pool4 in FCN-8s, the L-RNN is able to learn contextual information over a much larger range than the receptive field of pure local convolutions. "
This is potentially true, but I do not see why this was not also the case for FCN-32s (this is more a property of the recurrence rather than the 8/32 factor, right?)

A few additional points: 
It seems like Fig 2b and Fig2c never made it into the pdf. 

Figure 4 is unstructured and throws some 30 boxes to the reader - I would be surprised if anyone is able to get some information out of this (why not have a table?) 

Appendix A: this is very mysterious. Did you try other learning rate schedules? (e.g. polynomial)
What is the performance if you apply a standard training schedule? (e.g. step). 
Appendix C: "maps .. is" -> "maps ... are"

[Official Review · AnonReviewer3 · rating 5 · confidence 5 · 16 Dec 2016]
**No Title**

This paper proposes a cascade of paired (left/right, up/down) 1D RNNs as a module in CNNs in order to quickly add global context information to features without the need for stacking many convolutional layers. Experimental results are presented on image classification and semantic segmentation tasks.

Pros:
- The paper is very clear and easy to read.
- Enough details are given that the paper can likely be reproduced with or without source code.
- Using 1D RNNs inside CNNs is a topic that deserves more experimental exploration than what exists in the literature.

Cons (elaborated on below):
(1) Contributions relative to, e.g. Bell et al., are minor.
(2) Disappointed in the actual use of the proposed L-RNN module versus how it's sold in the intro.
(3) Classification experiments are not convincing.

(1,2): The introduction states w.r.t. Bell et al. "more substantial differences are two fold: first, we treat the L-RNN module as a general block, that can be inserted into any layer of a modern architecture, such as into a residual module. Second, we show (section 4) that the
L-RNN can be formulated to be inserted into a pre-trained FCN (by initializing with zero recurrence
matrices), and that the entire network can then be fine-tuned end-to-end."

I felt positive about these contributions after reading the intro, but then much less so after reading the experimental sections. Based on the first contribution ("general block that can be inserted into any layer"), I strongly expected to see the L-RNN block integrated throughout the CNN starting from near the input. However, the architectures for classification and segmentation only place the module towards the very end of the network. While not exactly the same as Bell et al. (there are many technical details that differ), it is close. The paper does not compare to the design from Bell et al. Is there any advantage to the proposed design? Or is it a variation that performs similarly? What happens if L-RNN is integrated earlier in the network, as suggested by the introduction?

The second difference is a bit more solid, but still does not rise to a 'substantive difference' in my view. Note that Bell et al. also integrate 1D RNNs into an ImageNet pretrained VGG-16 model. I do, however, think that the method of integration proposed in this paper (zero initialization) may be more elegant and does not require two-stage training by first freezing the lower layers and then later unfreezing them.

(3) I am generally skeptical of the utility of classification experiments on CIFAR-10 when presented in isolation (e.g., no results on ImageNet too). The issue is that CIFAR-10 is not interesting as a task unto itself *and* methods that work well on CIFAR-10 do not necessarily generalize to other tasks. ImageNet has been useful because, thus far, it produces features that generalize well to other tasks. Showing good results on ImageNet is much more likely to demonstrate a model that learns generalizable features. However, that is not even necessarily true, and ideally I would like to see that that a model that does well on ImageNet in fact transfers its benefit to at least one other ask (e.g., detection).

One additional issue with the CIFAR experiments is that I expect to see a direct comparison of models A-F with and without L-RNN. It is hard to understand from the presented results if L-RNN actually adds much. In sum, I have a hard time taking away any valuable information from the CIFAR experiments.

Minor suggestion:
- Figure 4 is hard to read. The pixelated rounded corners on the yellow boxes are distracting.

[Official Review · AnonReviewer1 · rating 7 · confidence 4 · 29 Dec 2016]
**Interesting approach to large field of view networks**

Please provide an evaluation of the quality, clarity, originality and significance of this work, including a list of its pros and cons.


Paper summary: this work proposes to use RNNs inside a convolutional network architecture as a complementary mechanism to propagate spatial information across the image. Promising results on classification and semantic labeling are reported.


Review summary:
The text is clear, the idea well describe, the experiments seem well constructed and do not overclaim. Overall it is not a earth shattering paper, but a good piece of incremental science.


Pros:
* Clear description
* Well built experiments
* Simple yet effective idea
* No overclaiming
* Detailed comparison with related work architectures


Cons:
* Idea somewhat incremental (e.g. can be seen as derivative from Bell 2016)
* Results are good, but do not improve over state of the art


Quality: the ideas are sound, experiments well built and analysed.


Clarity: easy to read, and mostly clear (but some relevant details left out, see comments below)


Originality: minor, this is a different combination of ideas well known.


Significance: seems like a good step forward in our quest to learn good practices to build neural networks for task X (here semantic labelling and classification).


Specific comments:
* Section 2.2 “we introduction more nonlinearities (through the convolutional layers and ...”. Convolutional layers are linear operators.
* Section 2.2, why exactly RNN cannot have pooling operators ? I do not see what would impede it.
* Section 3 “into the computational block”, which block ? Seems like a typo, please rephrase.
* Figure 2b and 2c not present ? Please fix figure or references to it.
* Maybe add a short description of GRU in the appendix, for completeness ?
* Section 5.1, last sentence. Not sure what is meant. The convolutions + relu and pooling in ResNet do provide non-linearities “between layers” too. Please clarify
* Section 5.2.1 (and appendix A), how is the learning rate increased and decreased ? Manually ? This is an important detail that should be made explicit. Is the learning rate schedule the same in all experiments of each table ? If there is a human in the loop, what is the variance in results between “two human schedulers” ?
* Section 5.2.1, last sentence; “we certainly have  a strong baseline”; the Pascal VOC12 for competition 6 reports 85.4 mIoU as best known results. So no, 64.4 is not “certainly strong”. Please tune down the statement.
* Section 5.2.3 Modules -> modules
* The results ignore any mention of increased memory usage or computation cost. This is not a small detail. Please add a discussion on the topic.
* Section 6 “adding multi-scale spatial” -> “adding spatial” (there is nothing inherently “multi” in the RNN)
* Section 6 Furthermoe -> Furthermore
* Appendix C, redundant with Figure 5 ?

[Author Response · Weidi Xie · 19 Jan 2017]
**Response to All Reviewers on Common Points**

Thank you very much for all the comments and suggestions.
Some of the comments were already addressed in version 2 of the paper.
Version 3 addresses most of the remaining points. 

We summarize the main changes here:
 
Version 3:
1. Added additional architectures for CIFAR-10 classification, including
-- Network E, F, where CNN modules and LRNN modules are interleaved at multiple levels. Network F outperforms the ResNet-164.
2. Reformulated Figure 4 (Architectures for CIFAR experiments) as Table 1,
3. Extended discussions on the experimental results.

Version 2: 
1. L-RNN modules with vanilla RNNs trained with Layer Normalization (Ba et al. 2016)
2. Baseline-CNN added containing only convolutional layers.

The following are the response to all reviewers on the common points:

---- All the reviewers comment on the very good paper by Bell et al. We address those points here.

Reviewer 1:
* Idea somewhat incremental (e.g. can be seen as derivative from Bell 2016).
 
Reviewer 2:
* Regarding novelty, the idea of combining a recurrent layer with a CNN, something practically very similar was proposed in Bell et al (2016). There are a few technical differences (e.g. cascading versus applying in parallel the recurrent layers), but in my understanding these are minor changes. The idea of initializing the recurrent network with the CNN is reasonable but is at the level of improving one wrong choice in the original work of Bell, rather than really proposing something novel.
 
Reviewer 3:
* Contributions relative to, e.g. Bell et al., are minor.

Response:
In our paper, the proposed L-RNN module aims to be general; it can be inserted at any stage in the architectures for capturing contextual information at multiple levels.

To be clear on the differences:

1. Interleaving CNN with L-RNN modules:
In Bell et al., the spatial RNNs are applied on top of CNN features (VGG-net) to learn contextual information at the final stage. A similar idea was also proposed in ReSeg (Visin et al. 2016).
Our paper goes further than Bell et al. or Visin et al. by interleaving the CNN and LRNN modules at multiple levels of the network.
In consequence, the network is capable of learning representations from both local and larger context at every layer, alleviating the limitations of a fixed kernel size.
 
2. How L-RNNs are inserted and initialized:
Under the scenario when the amount of data is limited, e.g. detection in Bell et al, semantic segmentation in our case.
In Bell et al., the authors take pre-trained CNN networks (VGG-16 up to conv5), and then train spatial RNNs on top of the pre-computed features.
In contrast, we show that a L-RNN can be directly inserted and trained within existing convolutional layers. This means that we increase the representational power of the pre-trained model directly. All layers in the pre-trained networks can therefore be adapted efficiently.

In our case, we re-purpose several layers in the pre-trained network as L-RNN modules at multiple levels, e.g. after pool3, pool4, and the final fully connected layers. As a result, we get a performance boost of 5% (mean IOU).
 
3. Choice of RNNs.
In Bell et al., to avoid computational cost, the authors choose to use ReLU-based vanilla RNN (rather than GRU or LSTM).
In our paper, we validate this choice by showing that vanilla RNN with Layer Normalization (Ba et al. 2016) achieves similar performance to GRU.
 
4. Separable Convolutions:
In Bell et al., 4 bidirectional spatial RNNs are applied to learn the global context. While in our paper, we propose to use 2, each bidirectional spatial RNNs is learning to approximate 1D convolutions, this idea comes from separable convolution.

---- Comments on CIFAR classification experiments.
 
Reviewer 2:
* Furthermore, as noted in my early question, Wide Residual Networks (Sergey Zagoruyko, Nikos Komodakis, BMVC16) report  better results on CIFAR-10 (4% error), while not using any recurrent layers (rather using instead a wide, VGG-type, ResNet variant). 
So, the authors answer: "Wide Residual Networks use the depth of the network to spread the receptive field across the entire image (DenseNet (Huang et al., 2016) similarly uses depth). Thus there is no need for recurrence within layers to capture contextual information. In contrast, we show that a shallow CNN, where the receptive field would be limited, can capture contextual information within the whole image if a L-RNN is used."
 
* So, we agree that WRN do not need recurrence - and can still do better.
The point of my question has practically been whether using a recurrent layer is really necessary; I can understand the answer as being "yes, if you want to keep your network shallow".  I do not necessarily see why one would want to keep one's network shallow.
 
Reviewer 3:
* One additional issue with the CIFAR experiments is that I expect to see a direct comparison of models A-F with and without L-RNN. It is hard to understand from the presented results if L-RNN actually adds much. In sum, I have a hard time taking away any valuable information from the CIFAR experiments.
 
Response:
Here is a short summary of the discussion part of the CIFAR experiments (now included in  the updated paper).

The L-RNN module is a general computational module; it is not our intention to replace the deep networks (e.g. residual variants). 

1. Why only use shallow networks?
It is well known that deep networks can generally achieve better results than shallow ones. Thus, in our experiments, we use relatively shallow networks to avoid the possibility that the performance gain is due to network depth. Our experiments confirm that increasing the network depth by only adding low-level CNN modules below a L-RNN improves the results (Table 2, Network A to D)

2. Interleaving L-RNN with CNN modules.
We show a comparison between a Baseline-CNN and Network-E in Table 2. Baseline-CNN is composed of 7 convolutional layers with 1.56M parameters, it achieves 8.48% top1 error. While the Network-E interleaved with CNN and L-RNN modules contains 0.97M parameters, achieving 5.96% top1 error. The difference between them is the added LRNN modules.
Moreover, by adding more layers, Network F(5.39% top1 error) achieves comparable performance to ResNet-164 (5.46%). 

---- Questions related to ImageNet.

Reviewer 2:
* Regarding the evaluation, experiments on CIFAR are interesting, but only as proof of concept.
* Probably an evaluation on imageNet would bring some more insight about the merit of this layer.
 
Reviewer 3:
* Classification experiments are not convincing.
* I am generally skeptical of the utility of classification experiments on CIFAR-10 when presented in isolation (e.g., no results on ImageNet too). The issue is that CIFAR-10 is not interesting as a task unto itself *and* methods that work well on CIFAR-10 do not necessarily generalize to other tasks. ImageNet has been useful because, thus far, it produces features that generalize well to other tasks. Showing good results on ImageNet is much more likely to demonstrate a model that learns generalizable features. However, that is not even necessarily true, and ideally I would like to see that that a model that does well on ImageNet in fact transfers its benefit to at least one other tasks (e.g., detection).
 
Response:
Further experiments on ImageNet are definitely on the top list of our future work.

---- Comments on Training Details.
 
Reviewer 1:
* Section 5.2.1 (and appendix A), how is the learning rate increased and decreased? Manually ? This is an important detail that should be made explicit. Is the learning rate schedule the same in all experiments of each table? If there is a human in the loop, what is the variance in results between “two human schedulers”?
 
Reviewer 2:
* Appendix A: this is very mysterious. Did you try other learning rate schedules? (e.g. polynomial)
 
Response:
The important message here is that restarting the learning rate several times can help the networks to escape saddle points or local minima.
We only experiment with the stepwise decay (no polynomial), and change the learning rate every 40, 60 or 80 epochs, we did not find much difference on this detail.
We only provide the intuitive explanation because it is not the main focus of this paper, it is just a training trick. If the readers are interested in this, we found two other related papers in this ICLR submission.

[Final Decision · Program Chairs · 06 Feb 2017]
**ICLR committee final decision**

This paper proposes a hybrid architecture that combines traditional CNN layers with separable RNN layers that quickly increase the receptive field of intermediate features. The paper demonstrates experiments on CIfar-10 and semantic segmentation, both by fine-tuning pretrained CNN models and by training them from scratch, showing numerical improvements. 
 
 The reviewers agreed that this paper presents a sound modification of standard CNN architectures in a clear, well-presented manner. They also highlighted the clear improvement of the manuscipt between the first draft and subsequent revisions. 
 However, they also agreed that the novelty of the approach is limited compared to recent works (e.g. Bell'16), despite acknowledging the multiple technical differences between the approaches. Another source of concern is the lack of large-scale experiments on imagenet, which would potentially elucidate the role of the proposed interleaved lrnn modules in the performance boost and demonstrate its usefulness to other tasks. 
 
 Based on these remarks, the AC recommends rejection of the current manuscript, and encourages the authors to resubmit the work once the large-scale experiments are completed.